# Operando Raman spectroscopy uncovers hydroxide and CO species enhance ethanol selectivity during pulsed $CO_2$ electroreduction

Antonia Herzog [1,2], Mauricio Lopez Luna[1,3], Hyo Sang Jeon[1,4], Clara Rettenmaier [1], Philipp Grosse[1], Arno Bergmann [1] & Beatriz Roldan Cuenya [1]

Pulsed $CO_2$ electroreduction ($CO_2RR$) has recently emerged as a facile way to in situ tune the product selectivity, in particular toward ethanol, without re-designing the catalytic system. However, in-depth mechanistic understanding requires comprehensive operando time-resolved studies to identify the kinetics and dynamics of the electrocatalytic interface. Here, we track the adsorbates and the catalyst state of pre-reduced $Cu_2O$ nanocubes (~30 nm) during pulsed $CO_2RR$ using sub-second time-resolved operando Raman spectroscopy. By screening a variety of product-steering pulse length conditions, we unravel the critical role of co-adsorbed OH and CO on the Cu surface next to the oxidative formation of $Cu-O_{ad}$ or $CuO_x/(OH)_y$ species, impacting the kinetics of CO adsorption and boosting the ethanol selectivity. However, a too low $OH_{ad}$ coverage following the formation of bulk-like $Cu_2O$ induces a significant increase in the $C_1$ selectivity, while a too high $OH_{ad}$ coverage poisons the surface for C-C coupling. Thus, we unveil the importance of co-adsorbed OH on the alcohol formation under $CO_2RR$ conditions and thereby, pave the way for improved catalyst design and operating conditions.

Within the scope of reducing global $CO_2$ emissions to limit climate change, carbon net-zero technologies have gained enormous interest. One promising technology is the $CO_2RR$, which closes the carbon cycle by using renewable energies to transform $CO_2$ back into useful chemicals and fuels[1]. Among the metals studied, only copper electrodes have the unique ability to produce the desired energy-dense alcohols and hydrocarbons such as ethanol and ethylene in significant amounts[2]. However, for further commercialization in high-current electrolyzers, Cu-based catalysts still suffer from a broad selectivity distribution, low activity, and low stability during long-term operation.

Several strategies have been developed to address these issues and to increase the product distribution toward $C_{2+}$ products, which include tuning the catalyst structure and composition[3–7] and modifying the electrolyte[8–10]. Additionally, the (initial) oxidation state of Cu plays a major role, particularly oxidized Cu species and oxide-derived Cu materials showed a major improvement in the selectivity and stability of the catalysts[11–16]. Thus, a simple way to regenerate the desired oxidation state of Cu in situ is by means of pulsed potential $CO_2RR$, where an electrocatalytic potential alternates between an oxidizing and a $CO_2RR$ potential[17]. Hereby, the key pulse parameters are the

[1]Department of Interface Science, Fritz Haber Institute of the Max-Planck Society, 14195 Berlin, Germany. [2]Present address: Massachusetts Institute of Technology, Research Laboratory of Electronics, 77 Massachusetts Ave, Cambridge, MA 02139, USA. [3]Present address: Chemical Sciences Division, Lawrence Berkeley National Laboratory, Berkeley, CA 94720, USA. [4]Present address: Korea Institute of Science and Technology, 5 Hwarang-ro 14-gil, Wolgok 2(i)-dong, Seongbuk-gu, Seoul, South Korea. ✉e-mail: abergmann@fhi-berlin.mpg.de; roldan@fhi-berlin.mpg.de

applied cathodic and anodic potential, the pulse shape as well as the pulse length[17].

By varying only the cathodic and anodic pulse lengths, while keeping the other pulse parameters constant, the catalytic properties of Cu electrodes could be enhanced[17–19]. In this way, the amount, as well as the type of Cu oxide species (Cu$^+$, Cu$^{2+}$) formed on the catalyst surface, was controlled[20,21]. In particular, distorted non-crystalline Cu(I)/Cu(II) domains were formed at short anodic pulses (below 2 s) as evidenced by operando X-ray absorption spectroscopy (XAS) and X-ray diffraction (XRD), which had been associated with an enhanced ethanol formation[18]. The dynamic alternation of the potential impacts also the restructuring (faceting and defects) and the roughening of the catalyst at longer anodic pulse lengths (above 1 s), which were observed to enhance the selectivity of either methane or ethylene[18,20,22]. However, the induced dynamics on the surface coverage of hydrogen (H$_{ad}$), hydroxide (OH$_{ad}$), and carbon monoxide (CO$_{ad}$) are crucial to fully understand the underlying principle of selectivity changes via pulsed CO$_2$RR. Several studies suggested that the application of the anodic pulse leads to higher OH coverage, while H$_{ad}$ is removed due to the positive polarization of the electrodes[19,21,23–26]. The resulting higher OH coverage was proposed to stabilize the active CO$_{atop}$ compared to the inactive CO$_{bridge}$ species[27–29], which then enhances the ethanol selectivity. The beneficial effect of a higher OH coverage and the resulting higher local pH was also linked to a change in the CO adsorption energy as well as a lower C–C coupling barrier[30]. While the CO binding configuration was previously tracked with operando surface-enhanced infrared absorption spectroscopy (SEIRAS)[27,29], there is to date no experimental evidence for the postulated changes in the OH and CO coverages, and the selectivity-determining role of the adsorbed OH beyond CO dimerization stays unclear. Moreover, it remains challenging to determine the possible presence of surface Cu oxides due to the lack of surface-sensitive operando characterizations[1]. Thus, these tools are needed to gain a better mechanistic comprehension of pulsed CO$_2$RR.

In this study, we, therefore, applied operando sub-second time-resolved surface-enhanced Raman spectroscopy (SERS), which is an ideal method to simultaneously track the changes in the surface oxidation state of Cu and the changes in the OH$_{ad}$ and CO$_{ad}$ surface adsorbates as well as their surface coverage. In this way, we followed the impact of the applied anodic or cathodic pulse lengths during pulsed CO$_2$RR. Thus, square-wave potential pulses were alternated between an electrocatalytic cathodic potential $E_c$ at −1.0 V and an oxidizing anodic potential $E_a$ at +0.6 V versus the reversible hydrogen electrode (RHE). Pre-reduced Cu$_2$O nanocubes (NCs) served as catalysts, which exhibit tunable selectivities for valuable C$_{2+}$ products such as ethanol or C$_1$ products such as methane, depending on the selected pulse length[18]. In particular, we correlated the enhancement of ethanol observed to an increase of the OH$_{ad}$ versus CO$_{ad}$ coverage and the formation of Cu-O$_{ad}$ and CuO$_x$/(OH)$_y$ species under selected pulsed CO$_2$RR conditions. Here, we demonstrate the impact of the adsorbates and surface species on the obtained selectivity trends in dependence on the applied pulse lengths to gain novel mechanistic insights in a sub-second time-resolved way.

## Results

### Time-dependent evolution of SERS

Figure 1a shows a scanning electron microscopy (SEM) image of the Cu$_2$O NC (~20 nm in size) pre-catalysts deposited on a glassy carbon electrode. The pre-catalyst was first pre-reduced in an electrochemical operando Raman flow cell setup (Supplementary Note 1, Supplementary Figs. 1, 2), resulting in loss of the initial cubic shape and the growth of the particles (~30 nm in size, Fig. 1b). Subsequently, pulsed CO$_2$RR with a selected cathodic pulse length $t_c$ and anodic pulse length $t_a$ was applied (Supplementary Table 1). For all pulsed CO$_2$RR measurements in this study, $E_c$ = −1.0 V and $E_a$ = +0.6 V (vs. RHE, for all shown

potentials in this study), and the electrolyte consists of 0.1 M CO$_2$-saturated potassium bicarbonate (KHCO$_3$) if not stated differently. We note that the morphology of the particles changed to larger structures after pulsed CO$_2$RR, as observed in the SEM image in Fig. 1c. Figure 1d presents the applied potential and the measured current (top) during the symmetric pulses with $t_c = t_a = 4$ s and the corresponding normalized SERS signal intensities over the Raman shift (bottom) as a function of the time. For the normalization of the SERS spectra, the intensity values were modified to a mean of 0 and a variance of 1. We note that the optical properties of the working catalyst are changing under potential pulse conditions as the reduced Cu$_2$O nanocubes transform into rough, porous nanostructures. Therefore, to account for different SERS enhancement at different pulse length conditions, the SERS bands in this study were only compared to each other within the same spectra to obtain intensity ratios (Supplementary Note 2). The periodic responses of the characteristic SERS bands in this spectroscopic region were fitted in Fig. 1e (with fit examples in Supplementary Fig. 3). The selected characteristic bands here are:

(I)   the CO$_{ad}$ vibrations on Cu, namely the Cu–CO rotation (CO$_r$) and stretching (CO$_s$) vibrations at 280 and 360 cm$^{-1}$, where CO is considered to be adsorbed on Cu as shown with surface characterization methods[31,32]. The CO bands appear during each cathodic pulse as CO$_2$RR intermediates and disappear during the anodic pulses[6,33,34]. The relative CO surface coverage (CO$_{cov}$) can be obtained by the intensity ratio of the two CO bands as CO$_{cov}$ = Intensity (CO$_s$)/Intensity (CO$_r$). This relationship was derived in detail in our previous study through operando measurements in the presence of different CO concentrations and through DFT vibrational analysis for different CO coverages on Cu(100) surfaces[33]. However, the structural and morphological differences between the Cu(100) derived relationship to the oxide-derived Cu used in this work do not allow a direct comparison of the ratios and the CO coverage:

(II)   the OH$_{ad}$ vibration on or close to Cu, which can appear during the cathodic pulse at 495 cm$^{-1}$ and during the anodic pulse redshifted at 450 cm$^{-1}$ if OH is adsorbed or close to the Cu surface via dipole-dipole interactions[34–36]:

(III)   the bands related to Cu$_2$O on the surface at 410 cm$^{-1}$ (multiphonon process), 530 cm$^{-1}$ (Raman-allowed $F_{2g}$ mode), and 620 cm$^{-1}$ (IR-allowed $F_{1u}$ mode), which appear during each anodic pulse due to the oxidation of Cu and disappear again during each cathodic pulse due to its reduction[6,34,35,37]. The additional presence of CuO or Cu(OH)$_2$ as a minor component cannot be excluded due to similar band positions[38].

While the assignment of CO$_{ad}$ and Cu$_2$O is straightforward, the assignment of OH$_{ad}$ is still under debate, and some groups also assigned the band at 495 cm$^{-1}$ to Cu–C and Cu–O species[35,39]. However, experiments in several previous studies with D$_2$O showed a shift of the band position, which supports this assignment to chemisorbed OH$_{ad}$ on Cu[35,36,40,41]. Furthermore, bands at ~500 and 700 cm$^{-1}$, detectable during the electrochemical oxidation of Cu(111) and polycrystalline Cu surfaces, were linked by other groups to surface OH species through D$_2$O experiments and DFT calculations[42]. Their DFT calculations propose that the 500 cm$^{-1}$ band corresponds to a top-site OH stretching mode, possibly aligning with our scenario. In contrast, the 700 cm$^{-1}$ band was attributed to the bending mode of free Cu–OH, a band mode that may be absent in our case due to co-adsorption with CO.

Additionally, a weak carbonate band at 1070 cm$^{-1}$ was observed in some of the SERS spectra during the first 20 s, as shown in Supplementary Fig. 4[33,35,43].

### Pulse length-dependent evolution of SERS

In order to follow the impact of the applied cathodic and anodic pulse duration on the catalyst structure and the adsorbates, we averaged the normalized operando SERS spectra of selected pulse conditions and

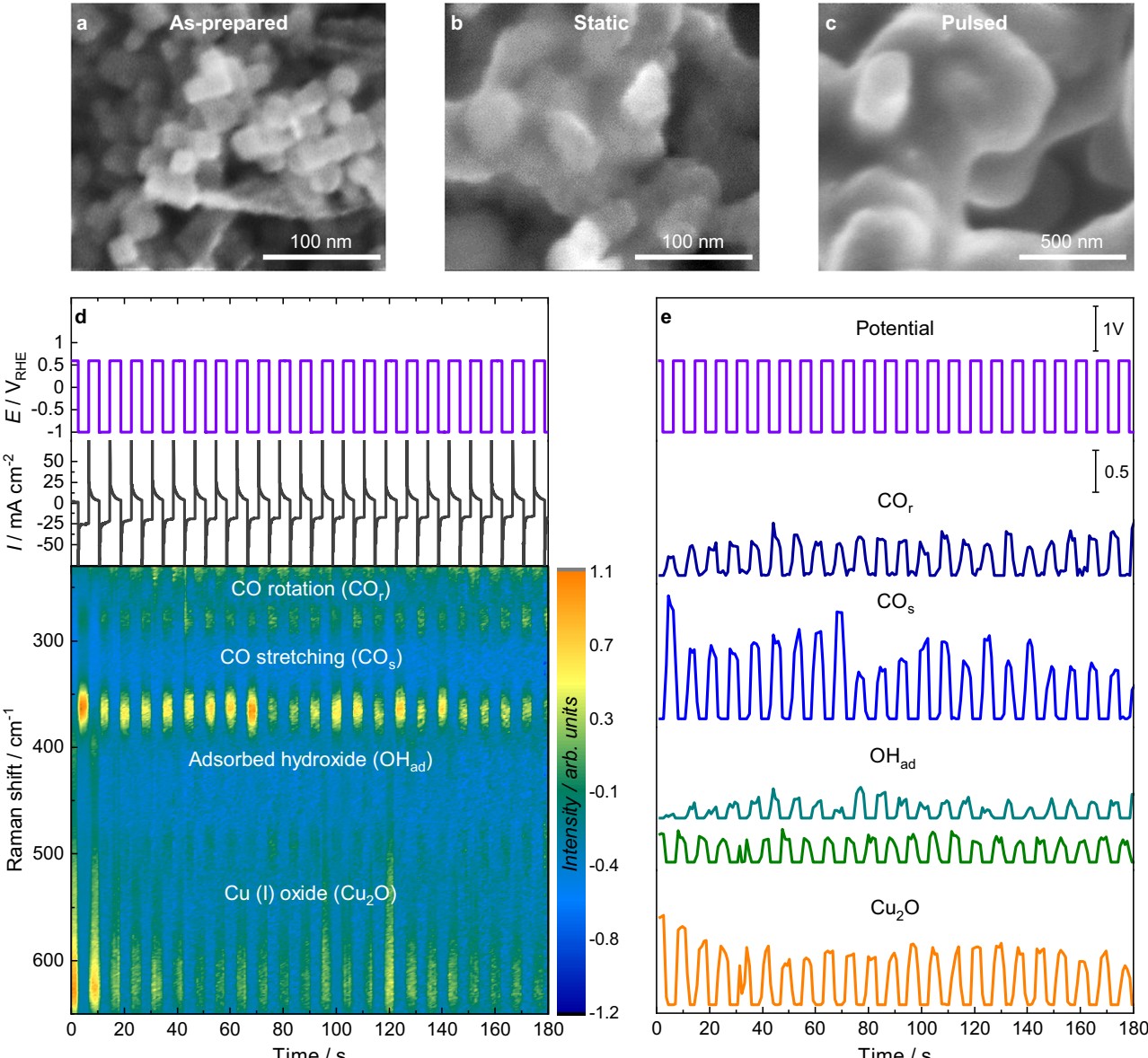

**Fig. 1 | Morphology and temporal evolution of the SERS intensity during pulsed $CO_2$RR. a** Ex situ SEM images of the as-prepared electrodes, **b** after potentiostatic $CO_2$RR at -1.0 V for 1 h, and **c** after pulsed $CO_2$RR at $E_c = -1.0$ V and $E_a = 0.6$ V with $t_c = t_a = 4$ s for 1 h. **d** Applied potential and current density over time with $t_c = t_a = 4$ s (top) and corresponding temporal evolution of the SERS signal intensity with marked Raman bands (bottom). **e** Applied potential over time and intensities of fits of selected characteristic Raman bands, namely the Cu–CO rotation ($CO_r$, 280 cm⁻¹, dark blue), Cu–CO stretching ($CO_s$, 360 cm⁻¹, light blue), Cu–$OH_{ad}$ ($OH_{ad}$, 495 cm⁻¹ at −1.0 V in turquoise and 450 cm⁻¹ at +0.6 V in green) and the copper(I) oxide ($Cu_2O$, sum of 530 and 620 cm⁻¹ divided by two, orange) bands.

fitted the characteristic bands over one pulse sequence, as shown in Fig. 2. The emphasis is placed on three pulse regimes that are typical for the highest ethanol (EtOH regime, Fig. 2a, d), ethylene/acetaldehyde ($C_2H_4O_x$ regime, Fig. 2b, e), and $C_1$ product selectivity ($C_1$ regime, Fig. 2c, f). The EtOH regime is characterized by short anodic pulses ($t_a = 0.5$ s), the $C_2H_4O_x$ regime by intermediate anodic pulses ($t_a = 4$ s), and in both cases, the cathodic pulses have an intermediate length of 4 s. In contrast, long anodic pulses ($t_a = 8$ s) combined with short cathodic pulses ($t_c = 0.5$ s) represent the $C_1$ regime[18]. Thereby, the intensity of the $CO_s$ compared to the $CO_r$ band decreased with shorter $t_a$ and is the smallest in the EtOH regime. The $OH_{ad}$ band is more intense during shorter $t_a$ (0.5 s, 4 s) in the EtOH/$C_2H_4O_x$ regime, while no $OH_{ad}$ bands, but stronger Cu oxide bands during long $t_a$ (8 s) and subsequent short $t_c$ (0.5 s) could be detected in the $C_1$ regime. These observations are underlined by additional SERS data at different pulse lengths (Supplementary Figs. 5, 6). During the anodic pulses, the $Cu_2O$ band intensities increased with the duration of the anodic pulse, being comparably weak at $t_a = 0.5$ s in the EtOH regime. Interestingly, there is also a contribution of Cu–$O_{ad}$ visible in the region of 606–626 cm⁻¹ during the anodic pulse in the EtOH regime for anodic pulses shorter than 2 s (Supplementary Fig. 5a, b)[42]. Under these conditions, also intense $OH_{ad}$ bands and additional bands at 370–380 and 520–540 cm⁻¹ were detected, which can be attributed to disordered $CuO_x/(OH)_y$ species[35,40]. Moreover, the carbonate band at 1070 cm⁻¹ was predominantly observed at the onset of both the anodic and cathodic pulses, possibly linked to dynamic alterations of the electrical double layer. However, due to the low band intensities under these conditions, it is challenging to thoroughly analyze its characteristics further (Supplementary Figs. 5–7). To verify the reproducibility of the data the experiments of the main product regimes were repeated on freshly prepared electrodes showing similar spectra and trends (Supplementary Fig. 9).

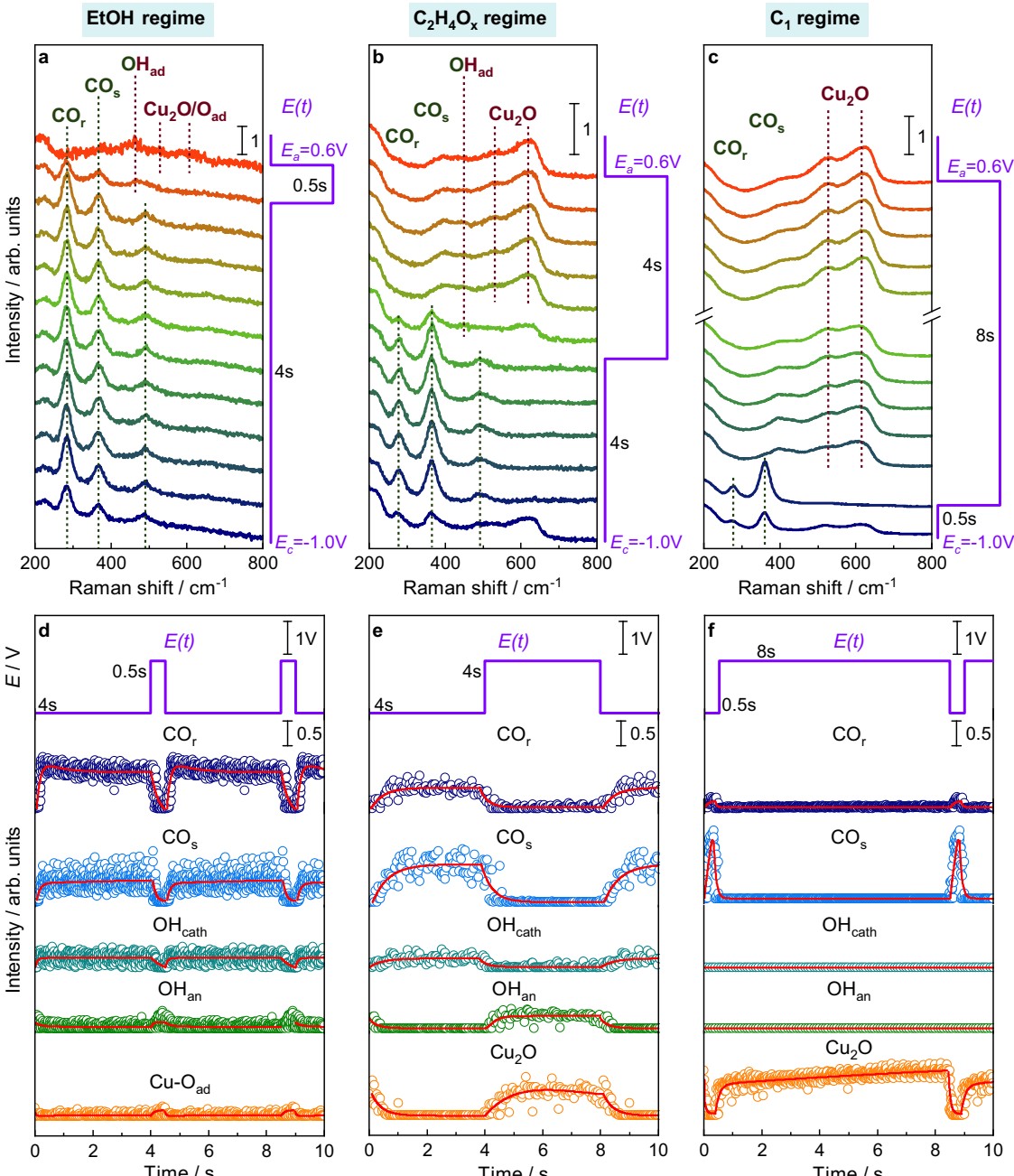

**Fig. 2 | Pulse length-dependent evolution of SERS spectra and characteristic bands. a–c** Normalized SERS spectra from bottom to top (as indicated with arrows) with highlighted characteristic SERS bands during pulsed $CO_2RR$ with varying pulse lengths at $E_c = -1.0$ V and $E_a = +0.6$ V and **d–f** intensities of fits of characteristic SERS bands averaged over one pulse sequence at selected pulse lengths. All averaged SERS spectra of **c** are shown in Supplementary Fig. 8. The data points in **d–f** represent the intensity fits of $CO_r$ (280 cm$^{-1}$, dark blue), $CO_s$ (360 cm$^{-1}$, light blue), $OH_{ad}$ (495 cm$^{-1}$ at $-1.0$ V in turquoise and 450 cm$^{-1}$ at $+0.6$ V in green), $Cu_2O$ (sum of 530 and 620 cm$^{-1}$ divided by two, orange) and Cu–$O_{ad}$ (610 cm$^{-1}$, orange) bands. The red lines denote the exponential fits and serve as guides for the eye.

To obtain additional insights into the average surface coverage of the characteristic species and their corresponding adsorption/desorption and oxidation/reduction kinetics, we performed exponential fits of the Raman band intensities during the averaged pulse profiles (Fig. 2d–f, Supplementary Fig. 10, Supplementary Tables 2–4). In particular, the kinetics of the ad- and desorption of CO on Cu altered with the applied pulse lengths (Supplementary Table 3). For example, in the EtOH regime, the $CO_s$ band intensity increased twice as fast during the cathodic pulse as it decreased during the anodic pulse. Additionally, the $CO_s/CO_r$ band intensity ratio, which reflects the CO surface coverage, gradually and persistently escalated throughout the cathodic pulse (Supplementary Fig. 11). These findings suggest a change in the

CO binding configuration during the cathodic pulse, which is expected to have implications for the catalytic function.

The differences in the CO binding configuration in terms of CO adsorption sites can be further evaluated by the C-O stretching vibration at higher wavenumbers (1900–2150 cm$^{-1}$, Supplementary Figs. 12, 13). Particularly, the SERS spectra show the contribution of bridge CO ($CO_{bridge}$ at 2030 cm$^{-1}$) and two atop CO ($CO_{atop}$) bands, namely the linear low-frequency CO ($CO_{LFB}$ at 2065 cm$^{-1}$) and high-frequency CO ($CO_{HFB}$ at 2095 cm$^{-1}$) bands, as described in the literature[29,44]. The contribution of $CO_{atop}$ sites, which are usually related to C–C coupling[44], is more prominent in the EtOH and $C_2H_4O_x$ regimes (Supplementary Fig. 13). Instead, the contribution of $CO_{bridge}$

sites, which are usually considered inactive for C–C coupling[27,29,44], increased significantly in the $C_1$ regime. Nevertheless, we found high uncertainties during several pulse sequences owing to the constantly evolving surface reactions that lead to the dynamic nature of the C–O stretching vibrations at high Raman shifts[34,44].

For the $OH_{ad}$ species, in turn, it is not easy to extract the ad- and desorption behavior since the bands additionally shifted upon the potential switch and partially overlapped with the evolving $Cu_2O$ bands during the anodic pulse (Fig. 2a, b). The shift of the bands probably resulted from the Stark effect due to the change in the electric field, with a reasonable Stark tuning rate of ~28 $cm^{-1}/V$[45], rather than from a change in the bonding configuration of $OH_{ad}$, as also discussed in the literature where the potential was only changed by 0.2 V[36]. Furthermore, it is likely that $OH_{ad}$ is an intermediate of the $Cu_2O$ formation during the anodic pulse.

The adsorption of oxygen on the Cu surface and the oxidation to $Cu_2O$ over the anodic pulse was, under all applied pulse conditions, slower than the corresponding desorption/reduction upon the application of the cathodic pulse (Supplementary Table 4). In the $C_1$ regime, the amount of surface $Cu_2O$ continuously increased the fastest, which indicates a continuous growth of the oxide layer over the course of the anodic pulse. Moreover, the spectroscopic weights of the three deconvoluted $Cu_2O$ bands changed at different applied pulse lengths (Supplementary Table 5). The relative intensity contribution of the band at 410 $cm^{-1}$ is stronger during longer cathodic pulses, which may have been influenced by the higher ratio of $OH_{ad}$[36]. On the other hand,

the intensity of the 530 and 620 $cm^{-1}$ bands contributes stronger for shorter cathodic pulses. Analyzing the time-dependent evolution of these bands in the $C_1$ regime (Supplementary Fig. 14) revealed that the relative intensity of the band at 410 $cm^{-1}$ was the highest in the first second of the anodic pulse, and its spectroscopic weight increased only slightly in the following. Also, the evolution of the intensity of the 620 $cm^{-1}$ band suggests a temporary maximum in the initial phase of the anodic pulse. In contrast, the intensity of the 530 $cm^{-1}$ band increased monotonously over the course of the anodic pulse. These changes might represent the formation/crystallization of the $Cu_2O$ layer terminating the Cu domains. In our previous work, we showed with operando XRD that over the course of an anodic potential pulse, crystalline $Cu_2O$ domains with a size of up to 3 nm were present at the end of a 10 s anodic pulse[18]. Additionally, operando XAS revealed a minor contribution of other Cu phases (e.g., Cu(II) compounds such as $CuO/Cu(OH)_2$) in the initial phase of the electrochemical oxidation, which are likely highly disordered and thus absent in the XRD data. However, they cannot be directly correlated to a specific cationic Cu (surface) species based on SERS.

## Correlations of selectivity, structure, and kinetics

To extract the link between the surface adsorbates and the catalytic function, Fig. 3 shows the change in the SERS band intensities (Supplementary Table 2) and in the Faradaic efficiencies ($\Delta$FEs) during pulsed $CO_2RR$ in comparison to static $CO_2RR$ conditions at −1.0 V obtained from a variety of anodic and cathodic pulse lengths

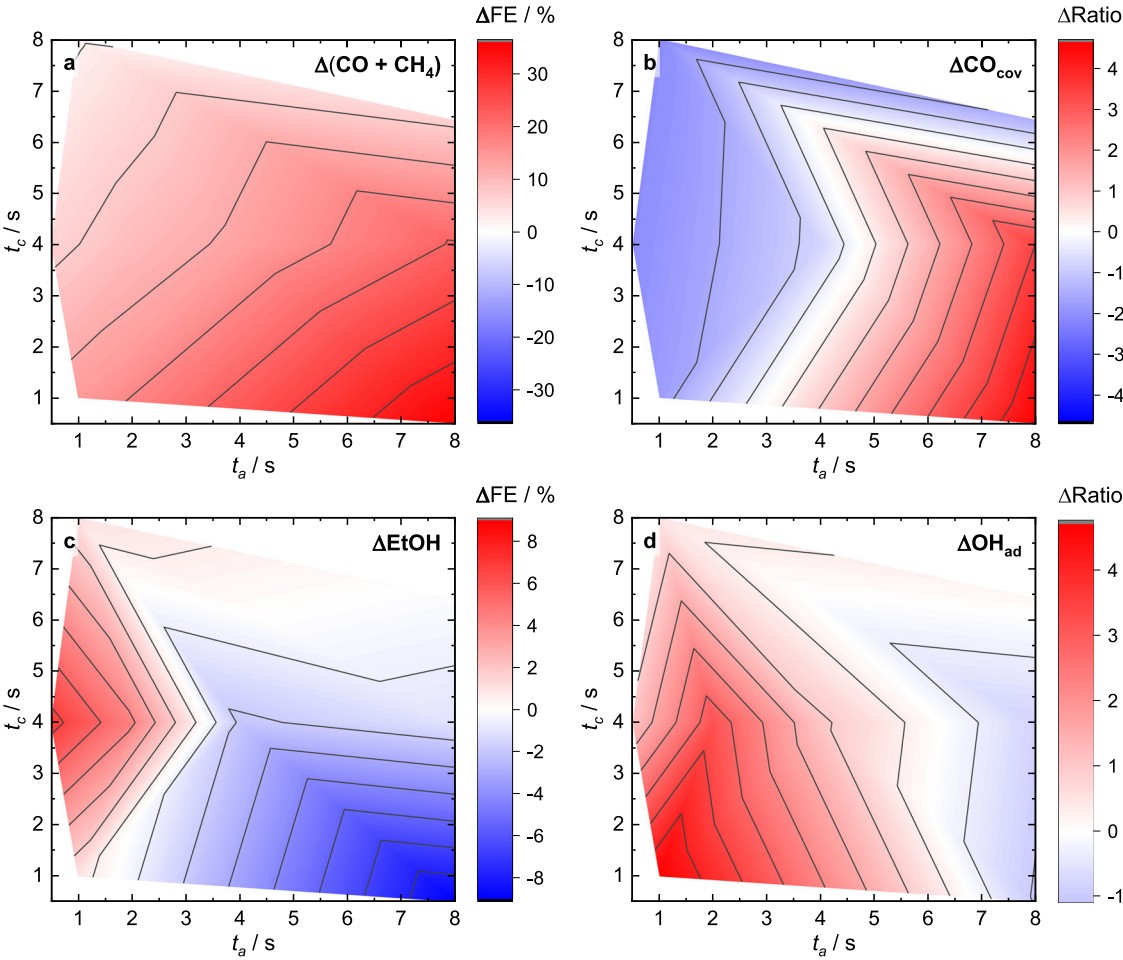

**Fig. 3 | Pulse length-dependent selectivities and adsorbates maps.** Changes in product selectivities during pulsed $CO_2RR$ with $E_c$ = −1.0 V, $E_a$ = +0.6 V with respect to the anodic and cathodic pulse lengths after subtraction of the corresponding values under static $CO_2RR$ conditions at −1.0 V (indicated with $\Delta$). **a** Change ($\Delta$) of the sum of Faradaic efficiencies ($\Delta$FEs) of the main $C_1$ products (CO and $CH_4$). **b** SERS intensity distribution of $\Delta$CO coverage on Cu. **c** $\Delta$FE of ethanol (EtOH). **d** Normalized SERS intensity distribution of $\Delta OH_{ad}$. The selectivity data were taken from identical catalysts included in our previous publication[18].

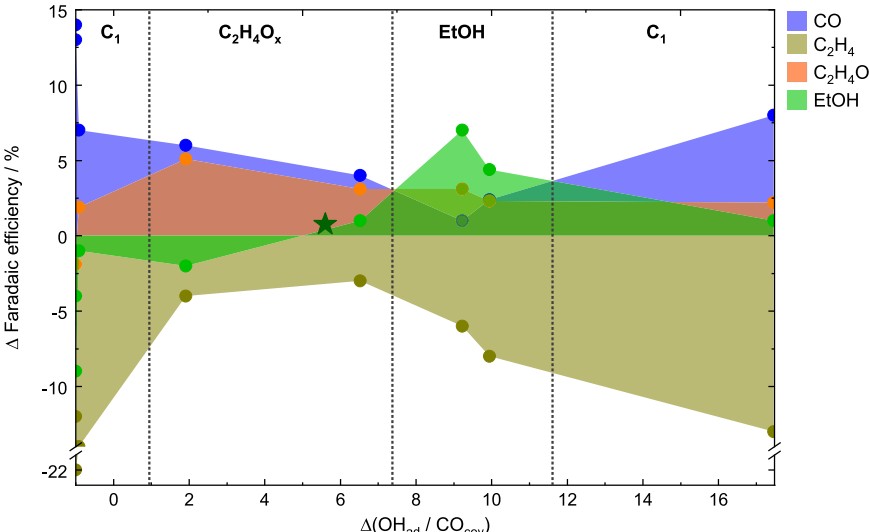

**Fig. 4 | Selectivity change correlated with $OH_{ad}$ / $CO_{cov}$ ratio.** Correlations between the selectivity change ΔFE of selected products (CO, ethylene, acetaldehyde, and ethanol) and the Δ ($OH_{ad}/CO_{cov}$) under pulsed $CO_2$RR conditions after subtracting the corresponding values under static $CO_2$RR conditions at −1.0 V. The green star represents the change of the ethanol selectivity during pulsed $CO_2$RR up to non-oxidizing potentials at $E_a = 0$ V.

conditions[18]. We depicted the ΔFEs rather than the partial current densities of the products to compare with the qualitative changes in the relative spectroscopic weight of adsorbate-related bands from SERS. Figure 3b depicts the changes in the $CO_{cov}$ during the cathodic part of pulsed $CO_2$RR, which could be determined by the ratio of the $CO_s$ and $CO_r$ band intensities[33]. The CO surface coverage has been previously shown to be important for the C-C coupling step, and a higher CO coverage could be linked to the increase of $C_{2+}$ products[33]. Under pulsed $CO_2$RR, the determined CO coverage is almost unchanged in the $C_2H_4O_x$ regime but decreased in the EtOH regime, while it increased in the $C_1$ regime compared to static $CO_2$RR conditions (Fig. 3a, b). Thus, the CO coverage does not seem to be the only crucial parameter determining the product selectivity under pulsed conditions.

Therefore, we further investigated the role of $OH_{ad}$, where its beneficial effect was so far only hypothesized or suggested from (theoretical) studies and/or not quantified[22,46,47]. Fig. 3d presents the change of the normalized $OH_{ad}$ intensities during the cathodic pulse (Supplementary Note 2). This map highlights the increased $OH_{ad}$ intensities at shorter anodic pulse lengths (<5 s) in comparison to static $CO_2$RR conditions, while the $OH_{ad}$ intensities decreased at longer anodic pulse lengths. This fits to the change of the $C_1$ products (Fig. 3a) for lower OH as well as to the enhanced EtOH selectivity at shorter anodic pulse lengths for higher OH intensities (i.e., coverage) (Fig. 3c).

In addition to the average band intensities, the kinetics of the intermediate formation are also expected to significantly influence the observed product generation due to the introduced variations by pulsing the potentials. Therefore, corresponding maps of the ad-/desorption of $CO_{ad}$ and $O_{ad}$, as well as the oxidation and reduction of $Cu_2O$ depending on the pulse lengths condition, could be created by the use of the time constant of the exponential fits from Fig. 2 (Supplementary Figs. 15–17, Supplementary Tables 3, 4). In the EtOH regime, $CO_r$ and $CO_s$ vibrations developed faster than they vanished (Supplementary Fig. 15); thus, CO was rapidly available to form EtOH at these pulse lengths. Furthermore, the kinetics of the ad- and desorption of $CO_s$ versus $CO_r$ (Supplementary Fig. 16), which directly impacts the surface CO coverage, indicate faster kinetics of $CO_s$ as compared to $CO_r$. This means that the rapidly available surface CO stems first mainly from $CO_s$, which has been related to the C-C coupling $CO_{atop}$ in the literature[33].

Moreover, even though we could not quantify the intensity of bands related to Cu oxides due their strong SERS sensitivity and missing normalization parameter, we can still follow the kinetic evolution of $O_{ad}$ at short anodic pulse lengths and surface $Cu_2O$ at long anodic pulse lengths (Supplementary Fig. 17). In particular, this demonstrates that Cu-$O_{ad}$ adsorbed kinetically quicker in the EtOH regime compared to the surface $Cu_2O$ in the $C_1$ regime, which could be crucial for the rapid availability of oxygen species to form EtOH.

## Hydroxide to carbon monoxide coverage ratio determines the products

To shift the focus from the applied pulse lengths, Fig. 4 directly plots the ΔFEs of selected products such as CO, $C_2H_4$, acetaldehyde and EtOH against the essential adsorbates previously identified, $OH_{ad}$ and CO, which are reflected here by the change of their ratio. The changes were calculated with respect to stationary reaction conditions. The corresponding plots showing all the other products can be found in Supplementary Fig. 18. Figure 4 shows that the FEs of $C_2H_4$ and $H_2$ under pulsed $CO_2$RR were lower, irrespective of the surface adsorbate composition, while the selectivities toward CO, acetaldehyde, and EtOH were (mostly) increased. In fact, Fig. 4 can be divided into four different product regions depending on the $OH_{ad}$/CO ratio. The lowest $OH_{ad}$/CO ratio (-1, where no $OH_{ad}$ is adsorbed) is characterized by the maximum of $C_1$ products, such as CO and $CH_4$. Increasing $OH_{ad}$/CO ratios (1–7.3) lead to the increased formation of $C_{2+}$ products such as $C_2H_4$ and acetaldehyde. A further increase of the $OH_{ad}$/CO ratios (7.3–11.6) correlates with the highest ΔFE of EtOH, together with other minor alcohols, while the ΔFEs of CO, $C_2H_4$, and acetaldehyde decreased, highlighting the beneficial effect of enhanced OH coverage for the ethanol formation. Interestingly, the improvement of the $CO_2$RR is not only seen in the FE of ethanol but also the partial current density increases at an optimal ratio of co-adsorbed CO and OH during $CO_2$RR (Supplementary Fig. 19). However, at even higher ratios of $OH_{ad}/CO_{cov}$ (>11.6) the FE of EtOH and ethylene started to decrease, while the FEs of $H_2$ and $C_1$ products such as CO and $COO^-$ (Supplementary Fig. 18a) increased. At these high ratios, the selectivities of EtOH and acetaldehyde were still slightly enhanced as compared to static $CO_2$RR conditions[18].

We note that the decrease of the ethylene FE may be predominantly linked to irreversible (morphological) catalyst changes

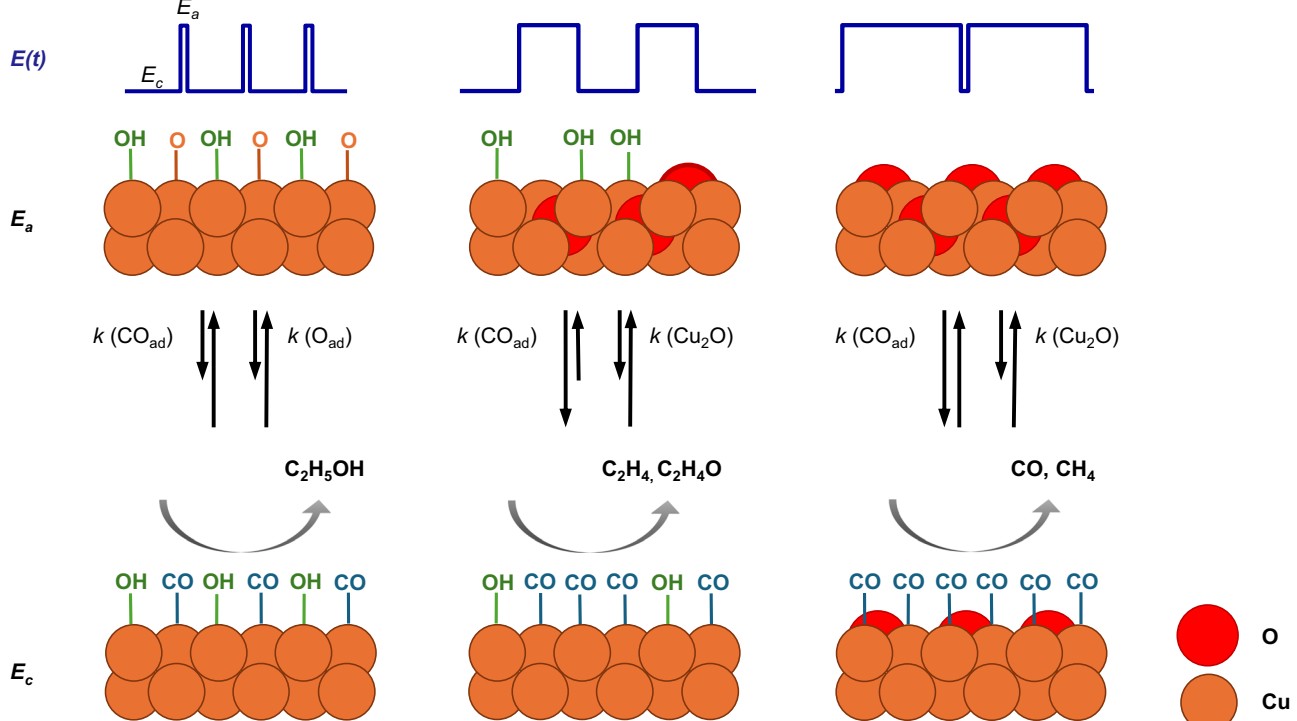

**Fig. 5 | Scheme of adsorbate and selectivity evolution during pulsed CO₂RR.** Depiction of the processes observed during pulsed CO₂RR with $E_c$ = −1.0 V, $E_a$ = +0.6 V of pre-reduced Cu₂O NCs in dependence of the anodic and cathodic pulse lengths. CO$_{ad}$, OH$_{ad}$, and O$_{ad}$ coverage are highlighted, and the oxidation state of Cu is indicated, which both contribute to different product selectivities during the cathodic pulse. The rate constant $k$ of the CO ad- and desorption, as well as the Cu reduction and oxidation, are schematically specified by the lengths of the arrows to each other, where a longer arrow is correlated to a faster process.

during the harsh working conditions, as shown in our previous work and will be further discussed later[18].

In order to verify the dependence of the EtOH selectivity on the OH$_{ad}$ concentration further, pulsed CO₂RR up to non-oxidizing $E_a$ = 0 V at $t_c/t_a$ = 4 s/1 s was also measured (Supplementary Fig. 20, Supplementary Table 6). The slight enhancement of EtOH compared to the static conditions goes along with a slight enhancement in OH$_{ad}$ versus CO$_{cov}$, as also indicated in Fig. 4 (green star), and supports the beneficial effect of OH$_{ad}$ for the EtOH production. These findings highlight the crucial role of OH$_{ad}$ and CO in the formation of EtOH.

## Discussion

From the results discussed, Fig. 5 suggests the mechanism for pulsed CO₂RR, where the periodic switching of the applied potential can change the Cu oxidation state and modify the adsorbate coverage of OH and CO with distinct kinetic behavior. Cu oxide-related species only formed during the anodic pulse and rapidly reduced again once the cathodic pulse was applied. At short anodic ($t_a$ < 2 s) and intermediate cathodic ($t_c$ = 4 s) pulse lengths, Cu-O$_{ad}$ vibrations and/or CuO$_x$/(OH)$_y$ species were detected, which also correlate well with the observation of Cu(II) at short anodic pulses from operando XAS data in our prior study[18]. These CuO$_x$/(OH)$_y$ species might be essential for the enhanced EtOH selectivity and are directly oxidized from Cu(0) to Cu(II) during the anodic pulse as no characteristic Cu₂O bands could be detected. Instead, at longer anodic ($t_a$ > 2 s) and shorter cathodic pulses ($t_c$ ≤ 4 s), typical Cu₂O bands appeared and reflected the growth and crystallization of Cu₂O over the course of the anodic pulse. Larger amounts of bulk-like Cu₂O, which were still present at the beginning of the cathodic pulse, led then to an enhancement in the methane selectivity linked to morphology changes, as also seen previously with XAS/XRD[22].

Most importantly, the Cu redox transitions, the OH surface coverage, and the local pH seem to be intertwined. In general, the

concentration of OH⁻ close to and adsorbed on the catalyst surface is expected to increase during pulsed CO₂RR at −1.0 V. Therein, OH⁻ species are produced by the reduction of H₂O during the cathodic pulse but are kept close to the catalyst surface due to the positive polarization of the Cu electrode during the anodic pulse. This is expected to increase the local pH during pulsed CO₂RR compared to static CO₂RR. The limited time at short $t_a$ values prevents the formation of an ordered Cu₂O phase and favors the formation of the detected Cu−O$_{ad}$ and/or CuO$_x$/(OH)$_y$ species. At longer oxidizing pulses, the low OH coverage values likely result from OH$_{ad}$/OH⁻ consumption for the Cu₂O formation (2 Cu + 2 OH⁻ → 2 Cu−OH → Cu₂O + H₂O)[22,24], leading to a decrease in the local pH during CO₂RR, which is known to favor methane and CO production[22]. However, too high coverages of OH$_{ad}$ start to block CO adsorption sites and prevent C−C coupling, which leads to an increase in C₁ products and HER[48].

It is important to note that the decrease of the ethylene and increase of the methane FE is linked to irreversible (morphological) catalyst changes caused by the pulsed reaction conditions (see ex-situ SEM images in Supplementary Fig. 21), which are especially prominent at longer anodic pulse length durations as also shown in our previous work[18]. Here, qualitative agreement to very high roughness and/or Cu dissolution of the Cu NCs could also lead to an increased population of low-coordinated sites, which are selective for hydrogen[49]. For example, sub-nanometer Cu clusters, likely formed under these continuous harsh redox cycles, were observed to be selective for methane formation[50,51]. This explains that under pulsed conditions, the highest CO coverage (without co-adsorbed OH$_{ad}$) was identified to enhance methane selectivities, while under less harsh stationary conditions, the highest CO coverage (usually with co-adsorbed OH$_{ad}$) was attributed to the highest C₂₊ product yield[6,33]. This suggests that under these pulsed reaction conditions, the C−C coupling is significantly hampered, leading to the reduction of the adsorbed CO to methane. To facilitate EtOH formation, the CO and OH coverage have to be balanced during

pulsed $CO_2RR$ to enhance the protonation of the $*CH_xCO$ intermediates and/or hamper its O removal[52], e.g., by a sufficient OH coverage blocking adjacent adsorption sites. Lower OH or higher CO coverages and, thus, more similar conditions as under static $CO_2RR$ shift the selectivity toward protonated $C_{2+}$ products such as ethylene and acetaldehyde. Therefore, we hypothesize that increasing the interfacial availability of base (OH) sites under (acidic) $CO_2RR$ conditions might lead to even higher EtOH formation.

In the EtOH regime, we found faster kinetics of CO adsorbate formation, which are expected to facilitate CO–CO dimerization and, thus, decrease $C_1$ product formation. The fast kinetics of the CO coverage in the EtOH regime might also be beneficial for the fast stabilization of $OH_{ad}$ since $CO_{ad}$ has already been demonstrated to stabilize $OH_{ad}$ by DFT calculations[46]. Furthermore, the fast kinetics of $Cu-O_{ad}$ in the EtOH regime compared to the formation of surface $Cu_2O$ in the $C_1$ regime during the anodic pulse demonstrate the rapid availability of oxygen species with short anodic pulse durations that are needed for ethanol formation.

In conclusion, this study revealed the link between the changes in the adsorbate structure and composition and the catalytic function of oxide-derived Cu nanocatalysts during pulsed $CO_2RR$ by utilizing time-resolved operando SERS. By the implementation of sub-second time resolution, the development of characteristic adsorbates such as OH and CO could be tracked during each individual pulse. In this way, the observed ethanol enhancement could be attributed to an optimal OH and CO surface coverage, which was found to be selectivity-determining towards alcohols over hydrocarbons. Furthermore, these OH species had a significantly lower concentration during stationary $CO_2RR$ and were not present in noticeable amounts on $Cu_2O$-derived working catalysts. Thus, only the intermittent formation of an OH/O-covered Cu surface triggers the continuous regeneration of the OH/CO-covered Cu catalyst during $CO_2RR$. Furthermore, the preferable CO ad-/desorption kinetics were found to contribute to higher ethanol yields. On the contrary, $OH_{ad}$ was found to be converted into bulk-like $Cu_2O$ species, which also led to the decrease of the near-surface pH and the formation of unfavorable $C_1$ products, such as methane and CO, as well as tremendously different reduction mechanisms, not being able to stabilize the OH species. All in all, this study underlines the urgency of time-resolved surface-sensitive techniques such as operando SERS to understand the reaction mechanism of pulsed $CO_2RR$ in order to favor the desired product selectivities. Furthermore, we finally confirmed experimentally the importance of surface base (OH) sites within the $CO_2RR$ mechanistic framework which will pave the road for in-depth investigations and novel catalyst design approaches.

## Methods
### Catalyst preparation
$Cu_2O$ NCs were synthesized by a ligand-free method, as described in our previous study[6]. The reagents were purchased from Sigma Aldrich in ACS grade and used without further purification. 5 mL of a $CuCl_2*2$ $H_2O$ solution (0.1 M) and 15 mL of a NaOH solution (0.2 M) were added to 200 mL of ultrapure water (>18 MΩ cm) at room temperature, and the solution was stirred for 5 min. Then, 10 mL of an L-ascorbic acid solution (0.1 M) was added to the mixture, and the solution was further stirred for 1 h. The solution was centrifuged and washed three times, twice with an ethanol-water mixture (1:1) and once with pure ethanol. The product was dried in a vacuum overnight, and the obtained powder was stored in the glove box.

To prepare the electrodes, 1 mg of the catalyst powder was dispersed in 0.5 mL of pure ethanol and ultrasonicated for 15 min to reach a concentration of $Cu_2O$ of 2 mg mL$^{-1}$. For the operando Raman measurements, 31 μL of the dispersion were drop-casted on one side of a polished glassy carbon electrode (8 × 8 mm, Glassy Carbon

SIGRADUR®, HTW) and dried at 60 °C for 5 min to obtain a $Cu_2O$ mass-loading of ~100 μg cm$^{-2}$.

### Electrolyte preparation
0.1 M $KHCO_3$ (Alfa Aesar, 99.7–100.5%) was purified with a cation-exchange resin (Chelex 100 Resin, Bio-Rad) and saturated with $CO_2$ (99.995%) for at least 15 min until a pH of 6.8 was reached.

### Operando surface-enhanced Raman spectroscopy
Operando SERS was performed with a Raman spectrometer (Renishaw, InVia Reflex) coupled with an optical microscope (Leica Microsystems, DM2500M) together with a motorized stage for sample tracking (Renishaw, MS300 encoded stage). Calibration of the system was carried out by using a Si(100) wafer (520.5 cm$^{-1}$). A near-infrared laser (Renishaw, RL785, $\lambda$ = 785 nm, $P_{max}$ = 500 mW, grating 1200 and 1800 lines mm$^{-1}$), as well as a HeNe laser (Renishaw, RL633, $\lambda$ = 633 nm, $P_{max}$ = 17 mW, grating 1800 lines mm$^{-1}$), were used as excitation sources. The backscattered light was Rayleigh-filtered and directed to a CCD detector (Renishaw, Centrus). For the operando measurements, the excitation source was focused on the surface of the sample, and Raman scattering signals were collected with a water immersion objective (Leica microsystems, ×63, numerical aperture of 0.9). The objective was protected from the electrolyte by a Teflon (FEP) film (Goodfellow, film thickness of 0.0125 mm), which was wrapped around the objective.

The electrochemical measurements were conducted at room temperature in a home-built spectro-electrochemical flow cell made of PEEK and controlled by a Biologic SP240 potentiostat (Supplementary Fig. 1). The cell was equipped with a leak-free Ag/AgCl reference electrode (LF-1-63, 1 mm OD, Innovative Instruments, Inc.) positioned close to the sample and a Pt counter electrode in the outlet of the flow. The working electrode with the catalyst drop-casted on glassy carbon was mounted from the bottom of the cell, and the area of the exposed catalyst was 0.25 mm$^2$. The electrolyte (0.1 M $KHCO_3$) was $CO_2$-saturated (pH = 6.8) in its reservoir ($V$ = 50 mL) outside of the Raman system and, from there, pumped through the cell with a peristaltic pump (PLP 380, Behr Labor-Technik).

The potentials in this manuscript were all converted to the RHE scale ($E$(vs. RHE) = $E$(vs. Ag/AgCl) + 0.242 V + 0.059 V × pH − iR) and corrected for $iR$ drop as determined by electrochemical impedance spectroscopy.

The collection time of each spectrum depends on the applied electrochemical protocol. For the pulsed $CO_2RR$ experiments at the cathodic potential $E_c$ = −1.0 V and the anodic potential $E_a$ = +0.6 V, acquisition times between 0.1 and 0.8 s were used, depending on the cathodic and anodic pulse lengths $t_c$ and $t_a$, respectively, to obtain at least three data points per pulse. The exact temporal resolutions for the low Raman shift region are given in Supplementary Table 1. To obtain a high time resolution, usually (if not stated differently), the static Raman mode in the region of 55–1272 cm$^{-1}$ was applied together with the 785 nm laser and the 1200 lines mm$^{-1}$ grating. In the region of 1700–2600 cm$^{-1}$, the 633 nm laser and the 1800 lines mm$^{-1}$ grating were used. The Raman data were first processed using the Renishaw WiRE 5.2 software to normalize the data and remove cosmic rays. Octave® scripts were written to combine the Raman and the electrochemical data, to fit characteristic Raman bands, and to average the Raman spectra. Averaged Raman spectra were obtained by averaging the Raman data points collected at the same times after the onset of each pulse cycle.

### Ex situ scanning electron microscopy
Ex situ SEM images of the samples prior to and after the different electrocatalytic conditions were measured using an SEM (Apreo SEM, Thermo Fisher Scientific) with an in-lens secondary electron detector. The samples were deposited on glassy carbon and directly rinsed with

ultrapure water (>18 MΩ cm) after each electrocatalytic $CO_2RR$ measurement to avoid electrolyte salt contamination on the sample.

## Selectivity measurements

The main part of the selectivity measurements was already published in our previous work[18] except for the new measurements acquired at $E_c = -1.0$ V, $E_a = 0$ V (versus RHE and $iR$ drop corrected) and $t_c = 4$ s, and $t_a = 1$ s for a total duration of 4000 s. The measurements were carried out in an H-type cell equipped with an anion-exchange membrane (Selemion AMV, AGC) separating the cathodic and the anodic compartments and controlled by an Autolab (Metrohm) potentiostat. A leak-free Ag/AgCl reference electrode (LF-1, Alvatek) served as the reference electrode, a platinum gauze electrode (MaTecK, 3600 mesh cm$^{-2}$) as the counter electrode, and $Cu_2O$ NCs deposited on carbon paper as the working electrode. The electrolyte was $CO_2$-saturated 0.1 M $KHCO_3$. Gas products were detected and quantified every 15 min by online gas chromatography (GC, Agilent 7890B), equipped with a thermal conductivity detector and a flame ionization detector. Liquid products were analyzed after each measurement with a high-performance liquid chromatograph (Shimadzu Prominence), equipped with a NUCLEOGEL SUGAR 810 column and a refractive index detector, and a liquid GC (Shimadzu 2010 plus), equipped with a fused silica capillary column and a flame ionization detector. The Faradaic efficiencies were calculated by taking only the cathodic part into account since there is no catalytic activity during the anodic pulse. Specifically, for the calculations of the Faradaic efficiencies of the gas products, the cathodic current was weighted by the term $\frac{t_c}{t_c + t_a}$ to correct for effective time under $CO_2RR$, while for the Faradaic efficiencies of the liquid products, just the cathodic charge was taken into account.

## Data availability

Additional SERS data, fitting parameter, time constants and additional $CO_2RR$ data are given in the Supplementary Information. The raw SERS data (which require specialized software to process) are available from the corresponding authors on request.

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

## Acknowledgements
This work was funded by the Deutsche Forschungsgemeinschaft (DFG, German Research Foundation), project no. 327886311-SPP 2080. Additional financial support from the European Research Council under grant ERC-OPERANDOCAT (ERC-725915) was also greatly appreciated. A.H. and C.R. acknowledge support by the IMPRS for Elementary Processes in Physical Chemistry. We also thank Petrik Bischoff for his help in designing and manufacturing the operando flow cell. We thank Dr. Chao Zhan (FHI) for helpful discussions. Open access funding was provided by the Max Planck Society.

## Author contributions
A.H., A.B. and B.R.C. contributed to writing the manuscript. A.B. and B.R.C supervised the study. A.H. designed, planned, and analyzed all operando SERS experiments. A.H. and M.L.L. designed the operando SERS cell setup. A.H. and H.S.J. prepared the $Cu_2O$ NC samples. A.H. and C.R. performed the electrocatalytic $CO_2$RR selectivity measurements. P.G. carried out the SEM measurements.

## Funding

## Competing interests
The authors declare no competing interests.
