## [Peer Review File · Nature Communications]

REVIEWER COMMENTS

Reviewer #1 (Remarks to the Author):

In this work, Herzog et al. systematically explored how the relative surface coverage of OH and CO are controlled by pulsed electroreduction of CO₂. In particular, they probed how the length of the anodic pulse affects the relative coverages. They correlated those changes to the observed changes in Faradaic efficiency. This work demonstrates that the CO coverage does not solely describe the selectivity changes. Rather, the ratio of hydroxide to CO coverage appears to be a more robust descriptor of catalytic selectivity. I estimate that this observation is of great interest to the broad field of CO₂ reduction catalysis. That adsorbed OH may play an important role in CO₂ reduction catalysis has been hypothesized before. This work experimentally confirms this hypothesis. I therefore consider it a significant advance in the field.

The manuscript is generally well written and the experiments and methods of analysis appear robust. I therefore recommend publication of the manuscript after minor revision. I suggest the authors address the following points:

(1) Figure 4 (and the discussion in the text) focuses on changes in selectivity as a function of anodic pulse length. Although I find the compiled information very appealing and informative, the Faradaic efficiency does not inform about the partial currents (or rates of formation) for individual products. Correlating the ratio of the coverage with the rates would be more interesting. The change in Faradaic efficiency for hydrogen evolution is not shown in Figure 4. So the Faradaic efficiency somewhat obscures the origin of the selectivity change (is it because a particular product forms faster or is it because another product, such as H₂, forms more slowly?). Could the authors comment on why they focus on Faradaic efficiencies rather than on partial currents?

(2) On p. 6 of the manuscript, the authors note that they averaged the “normalized operando SERS spectra”. At this point in the manuscript, it is not explained what “normalized” means in this context. There is a brief note in this regard in the SI (Supplementary Note 2), but it appears that it is not referenced here. I suggest that the authors move supplementary note 2 to the main text. When I read the text, I was wondering how changes in the SERS enhancement factor are taken into account. It will be helpful to have this information early in the main text.

(3) Minor comments: Figure 1d: The dimensions of current should be mA/cm² not mA/cm; Supplementary Note 2: The word “absolute” is misspelled. Methods section, p. 17: Two references are not displayed.

Reviewer #2 (Remarks to the Author):

The revised work presents a deep insight into the pulsed CO₂ electroreduction mechanism as elucidated using operando Raman spectroscopy. By extracting some key information from operando Raman spectroscopy, the authors of this work have revealed the crucial role of OH- and CO adsorbed on the electrode surface in the selectivity of the electrode towards specific C₁ or C₂ products. In general, the pulsed conditions trigger the generation of the OH/CO-covered Cu that is responsible for the higher selectivity towards ethanol. On the other hand, at extremely high concentrations of OH respect to CO on the electrode surface, the selectivity is moved into mostly C₁ products such as CO or methane. In general, the work highlights the importance of combining electrochemical measurements with high quality spectroscopic response, in order to better understand the electrochemical interface that drive the performance of the catalyst towards a specific product. Overall, the work was well executed, and the experimental design was very well planned with a clever approach to relate the spectroscopic data to the experimental electrochemical results. In spite of that, I have some questions and concerns about the work that I would like to clarify. I believe the manuscript needs some careful evaluation to address the above comments, prior it could be considered for publication in Nature Communications.

1. In the following expression “the OHad vibration on Cu, which can appear during the cathodic pulse at 495 cm⁻¹ and during the anodic pulse redshifted at 450 cm⁻¹ if OH is adsorbed or close to the Cu surface” it is argued that these are signals related to OH adsorbed on Cu surface, however, some other authors support in literature that this signal should be located at 680 - 708 cm⁻¹ (J. Am. Chem. Soc. 2019, 141, 31, 12192–12196). Since this is a critical point in this article and also has some discussion in literature with controversial data. Have the authors done some isotopic labelling experiment to further support that this signal really come from OH? If not, it is recommended to provide such experiment to fully identify and better relate the data.

2. Regarding the CO_{cov}, although authors refer to a recent literature, for clarifying to the potential readers, I think some extra information about how to estimate/calculate CO_{cov} from the Raman data, including the proper equation and the method to calculate it should be provided.

3. I am concern about the reproducibility of the data. As it is well known, when working with plasmonic materials, there is the possibility of placing on/or nearby a hot-spot, which enhance the signal allowing to better follow the changes in Raman signal. A common practice when working in this field and, in order to give representative measurements, providing a similar behaviour in

different parts of the electrode is recommended. Have the authors made sure that the signals are representative along the studied electrode? Could the authors provide some reproducibility data?

4. In page 7, authors argue that “For the OHad species, in turn, it is not easy to extract the ad and desorption behavior since the bands additionally shifted upon the potential switch”. Although authors argue this, they do not show the extend for the change in the Raman shift. It is recommended to provide a plot with the change in the Raman shift to further support that the calculation of the ad and desorption is difficult to calculate. Related to this, why the shifting makes it difficult to calculate the ad and desorption rate constant?

5. In the statement “If one follows the time-dependent evolution of these bands in the C1 regime (Supplementary Fig. 13), the contribution of the band at 410 cm^{-1} was the highest in the first second...” This is not clearly shown in supplementary Fig. 13, could the authors clarify this point?

6. Page 12, line 283, for me, an increase in COO is not clear from the figure. Could authors clarify this point?

7. My concern about data shown in Figure 3 is related to the validity of the SERS values. It is well known that the intensity of a specific band depends not only on the quantity but also on the SERS enhancement at this specific moment that depends on the changes in the substrate. In other words, we should discard the reason for the value of the signals, whether they come from the amount of material or on the SERS enhancement. The authors explain very well that they used the COcov (from the ratio between COs/COR) to normalize the data. This is a very clever method; however, I think the validity of such normalization should be demonstrated. I would think that, in order to normalize the data, a spectator compound, which is a compound that is not participating in the reaction as an intermediate or something similar, should be more ideal. In this case, if the spectator increases the signal because of a SERS effect, this signal would serve as a normalization parameter (as it happens with an internal standard in analysis). Therefore, I am not sure about the validity of using COcov as a normalization parameter. I think authors should address this point since, from this data, most of conclusions of the authors are given in this work, so I think this is the key point of the article. I strongly recommend to include a demonstration of the validity of using COcov as such normalization parameter.

Minor changes

1. In the supporting info, Supplementary Figure 7 and Supplementary Figure 10 are missing, so I lose some information when evaluating this data.

2. Page. 9, line 216, maybe authors refer to Supplementary Table 2 instead of Table 1?

3. In Supplementary Figure 4, 5, 6, and 11 there is no sense in including “Y axis” values. They should be similar to those in the main text of the manuscript.

4. Page 11. Line 254, correct the expression “Cu-oxide related bands due their strong”

Reviewer #3 (Remarks to the Author):

The manuscript by Antonia Herzog and co-worker presents an extensively experimental study on pulsed CO₂ electroreduction to C₁ and C₂ species on copper electrodes. The authors provided very rich data to support their conclusion of the role of hydroxide surface coverages related to the faradaic efficiencies of different products. This should be a very interesting and important work to modulate the surface coverage to tuner the reaction products. I think it should be proper for publication on the Journal after they consider questions and comments as follows.

1. In this work the surface-enhanced Raman spectroscopy is used here, but the authors didn't provide any characterization of surface plasmon resonance in the present copper electrodes. Could they provide the related data for the optical property of the present electrode materials?

2. Page 7, lines 161-181, the authors discuss the low vibrational frequencies and higher vibrational frequencies. These Raman spectral peaks are contributed to the different adsorption site and C-C coupling. On the other hand, there is another possibility, which the Raman peaks arise from the dipole-dipole coupling, as discussed in previous studies. So it is better to provide further interpretation on this point.

3. Page 9, line 209, in the sentence “These changes might represent the formation/crystallization of the Cu₂O layer”, how to understand the surface can be thought as the crystallization of the Cu₂O layer? Could the authors provide further evidence?

4. Page 11, lines 248-252, when the discuss the kinetics of the adsorption and desorption of COs and CO_r vibrations, they thought the surface coverage of adsorbed CO, but it is not clear what/how much is the surface coverage of adsorbed CO on the copper surface?

5. Page 11, line 262, Figure 4 and Supplementary Figure 17, these authors discuss the influence of OH adsorption on the change of Faradaic efficiencies different reduced species. It is quite strange why ethylene has so large change in this Faradaic efficiency in the wide range of OHad/Coad ratio? This becomes especially significant in the lower and higher regimes. They should present further comments for this point.

6. In the reference list, line 607, reference 45 should be corrected.

REVIEWER REPORTS:

Reviewer #1:

In this work, Herzog et al. systematically explored how the relative surface coverage of OH and CO are controlled by pulsed electroreduction of CO₂. In particular, they probed how the length of the anodic pulse affects the relative coverages. They correlated those changes to the observed changes in Faradaic efficiency. This work demonstrates that the CO coverage does not solely describe the selectivity changes. Rather, the ratio of hydroxide to CO coverage appears to be a more robust descriptor of catalytic selectivity. I estimate that this observation is of great interest to the broad field of CO₂ reduction catalysis. That adsorbed OH may play an important role in CO₂ reduction catalysis has been hypothesized before. This work experimentally confirms this hypothesis. I therefore consider it a significant advance in the field.

The manuscript is generally well written and the experiments and methods of analysis appear robust. I therefore recommend publication of the manuscript after minor revision. I suggest the authors address the following points:

Response: We are very grateful for the time that the reviewer invested into the careful evaluation of the manuscript as well as for the positive comments.

(1) Figure 4 (and the discussion in the text) focuses on changes in selectivity as a function of anodic pulse length. Although I find the compiled information very appealing and informative, the Faradaic efficiency does not inform about the partial currents (or rates of formation) for individual products. Correlating the ratio of the coverage with the rates would be more interesting. The change in Faradaic efficiency for hydrogen evolution is not shown in Figure 4. So the Faradaic efficiency somewhat obscures the origin of the selectivity change (is it because a particular product forms faster or is it because another product, such as H₂, forms more slowly?). Could the authors comment on why they focus on Faradaic efficiencies rather than on partial currents?

Response: We agree to the reviewer that the FE alone does not provide the full picture of the product formation. In this work, we basically followed our catalytic insights from the recent paper in Nature Catalysis [REF 18] as we now studied the Cu₂O nanocubes under pulsed CO₂RR using operando Raman spectroscopy. In our previous work, we also showed a map with the total current density under a variety of pulsed CO₂RR conditions (Figure R1). Therein, it is clear that the pulsed CO₂RR mainly decreases the current density and thus, the rate of product formation. However, the highest current densities which are comparable to the stationary CO₂RR conditions were obtained in the upper left corner i.e. for short anodic and longer cathodic pulses. It is important to note that these conditions also include the preferable conditions for EtOH formation. We followed the reviewer's suggestion and added Supplementary Figure 19, which shows the change in partial current densities instead of FEs in Figure 4. This figure highlights an increase in the partial current densities of ethanol at the optimal ratio of co-adsorbed CO and OH during CO₂RR similar to the increase of the ethanol FE.

We are still keeping the FEs in the main discussion as in our opinion, we should not speculate by comparing the relative changes in surface coverage of adsorbates with the reaction rates. Using operando Raman spectroscopy, we have basically only access to changes in the relative spectroscopic weight of adsorbate-related bands and a direct quantitative link to the surface

coverage is not possible, at least in the absence of additional theoretical support. Here, we are missing procedures to calibrate the CO_s/CO_f ratio to realistic nanoparticulate Cu and thus, refrained from too speculative statements. The reaction rate and (partial) current densities however are strongly correlated to the number of active sites or the electrochemically active surface area on the electrode. But these are subject to rapid changes under the pulsed CO_2RR similar to the total current density. Thus, a proper comparison would require an on-the-fly determination of the evolving electrochemical surface area that we used for normalization of the (partial) current density during pulsed CO_2RR , which is not possible *inter alia* due to the contribution of the (porous) carbon substrate.

To clarify our approach, we added an explanation to the corresponding section, when introducing the Faradaic efficiencies in the manuscript.

Regarding the changes in the H_2 formation, we would like to refer the reviewer to Supplementary Fig. 18 in which we show that the H_2 FE is almost independently of the pulsed conditions decreased by $\sim 10\%$ points.

Figure R1: The dependency of the total current under pulsed CO_2RR conditions on the pulse lengths t_a and t_c . The values for the cathodic and anodic potentials are $E_c = -1.0\text{ V}$ and $E_a = 0.6\text{ V}$ vs. RHE, respectively. Taken from REF 18.

Changes to the manuscript:

Page 10, Line 265. “We depicted the ΔFEs rather than the partial current densities of the products to compare with the qualitative changes in the relative spectroscopic weight of adsorbate-related bands from SERS.”

Page 13, Line 334. “Interestingly, the improvement of the CO_2RR is not only seen in the FE of ethanol but also the partial current density increases at an optimal ratio of co-adsorbed CO and OH during CO_2RR (Supplementary Fig. 19).”

Changes to the supplementary manuscript:

Page 19. Addition of Supplementary Figure 19.

Supplementary Figure 19. Correlations between the change of the current density J of selected products (CO, ethylene, acetaldehyde, and ethanol) and the $\Delta(\text{OH}_{\text{ad}} / \text{CO}_{\text{cov}})$ under pulsed CO_2RR conditions after subtracting the corresponding values under static CO_2RR conditions at -1.0 V. The green star represents the change of ethanol selectivity during pulsed CO_2RR up to non-oxidizing potentials at $E_a = 0$ V. The current densities were taken from our previous work.⁵

(2) On p. 6 of the manuscript, the authors note that they averaged the “normalized operando SERS spectra”. At this point in the manuscript, it is not explained what “normalized” means in this context. There is a brief note in this regard in the SI (Supplementary Note 2), but it appears that it is not referenced here. I suggest that the authors move supplementary note 2 to the main text. When I read the text, I was wondering how changes in the SERS enhancement factor are taken into account. It will be helpful to have this information early in the main text.

Response: We express our gratitude to the reviewer for bringing this to our attention. It is crucial for the reader to comprehend the consideration of surface enhancement. Consequently, we have relocated a section from the method part and details from Supplementary Note 2 to the initial part of the manuscript.

Changes in the manuscript:

Page 4, Line 102. “For the normalization of the SERS spectra, the intensity values were modified to a mean of 0 and a variance of 1. [...]. Therefore, to account for different SERS enhancements at different pulse length conditions the SERS bands in this study were only compared to each other within the same spectra to obtain intensity ratios (Supplementary Note 2).”

(3) Minor comments: Figure 1d: The dimensions of current should be mA/cm² not mA/cm; Supplementary Note 2: The word “absolute” is misspelled. Methods section, p. 17: Two references are not displayed.

Response: We corrected these minor comments in the manuscript and supplementary manuscript.

Changes in the manuscript:

Page 6, Figure 1d. Correction of “mA cm⁻¹” to “mA cm⁻²”.

Page 18, Line 465. Correction of formatting error to “Supplementary Fig. 1”.

Page 18, Line 476. Correction of formatting error to “Supplementary Table 1”.

Changes in the supplementary manuscript:

Page 2, Line 37: Correction of “aboslute” to “absolute”.

Reviewer #2 (Remarks to the Author):

The revised work presents a deep insight into the pulsed CO₂ electroreduction mechanism as elucidated using operando Raman spectroscopy. By extracting some key information from operando Raman spectroscopy, the authors of this work have revealed the crucial role of OH- and CO adsorbed on the electrode surface in the selectivity of the electrode towards specific C₁ or C₂ products. In general, the pulsed conditions trigger the generation of the OH/CO-covered Cu that is responsible for the higher selectivity towards ethanol. On the other hand, at extremely high concentrations of OH respect to CO on the electrode surface, the selectivity is moved into mostly C₁ products such as CO or methane. In general, the work highlights the importance of combining electrochemical measurements with high quality spectroscopic response, in order to better understand the electrochemical interface that drive the performance of the catalyst towards a specific product. Overall, the work was well executed, and the experimental design was very well planned with a clever approach to relate the spectroscopic data to the experimental electrochemical results. In spite of that, I have some questions and concerns about the work that I would like to clarify. I believe the manuscript needs some careful evaluation to address the above comments, prior it could be considered for publication in Nature Communications.

Response: We appreciate the reviewer's time dedicated to the thorough evaluation of the manuscript, along with the positive comments and constructive criticism that we have used to improve the revised manuscript.

1. In the following expression “the OHad vibration on Cu, which can appear during the cathodic pulse at 495 cm⁻¹ and during the anodic pulse redshifted at 450 cm⁻¹ if OH is adsorbed or close to the Cu surface” it is argued that these are signals related to OH adsorbed on Cu surface, however, some other authors support in literature that this signal should be located at 680 - 708 cm⁻¹ (J. Am. Chem. Soc. 2019, 141, 31, 12192–12196). Since this is a critical point in this article and also has some discussion in literature with controversial data. Have the authors done some isotopic labelling experiment to further support that this signal really come from OH? If not, it is recommended to provide such experiment to fully identify and better relate the data.

Response: We agree to the reviewer's comment that the assignment of the OH adsorbate is crucial for our study. We had in fact already mentioned in the original submission that there is some controversy on the band assignment in the literature. Nonetheless, prompted by the reviewer and by doing a further extensive literature research we still think that the majority of the literature works, especially those who have performed isotopic labeling experiments with heavy water, support the OH assignment.

<https://chemistry-europe.onlinelibrary.wiley.com/doi/10.1002/celec.202001598>

<https://www.sciencedirect.com/science/article/pii/S0013468600004345?via%3Dihub>

<https://pubs.acs.org/doi/full/10.1021/jacs.0c02354>

<https://pubs.acs.org/doi/full/10.1021/jp983787c>

Referring to the mentioned JACS work (J. Am. Chem. Soc. 2019, 141, 31, 12192–12196), we should highlight the differences in the reaction conditions therein. In contrast to our work, the early stages of the electrochemical oxidation of Cu(111) were studied in comparably alkaline and CO₂/carbonate-free conditions (pH 12). Thus, no co-adsorption with CO is expected which can change the adsorbates configuration and thus, the molecular vibrations on the surface. The complexity of adsorbate-adsorbate interactions are also contributing to the broadness of the Raman bands in this region. It has been nicely shown that the OH peak position is substantially influenced by the interaction of the adsorbed OH with co-adsorbed C-related species. (Moradzaman, Mul, *ChemElectroChem* 2021, 8, 1–9. 10.1002/celec.202001598). We have to note that various binding configurations and vibrational models of adsorbed OH are listed in the SI of the JACS work mentioned by the reviewer. These could give rise to the Raman bands in the 430 to 503 cm⁻¹ region such as the Cu-OH stretching band on Cu(111) at 459 cm⁻¹, which may correspond to the band mode in our case, as the surface is covered by various adsorbates e.g. CO. Therefore, we have now extended the discussion of the peak assignment to provide the reader a better chance to judge the validity of our assignment.

Changes in the manuscript:

Page 6, Line 159. “However, experiments in several previous studies with D₂O showed a shift of the band position, which supports this assignment to chemisorbed OH_{ad} on Cu.^{35, 36, 40, 41} Furthermore, bands at ~500 and 700 cm⁻¹, detectable during the electrochemical oxidation of Cu(111) and polycrystalline Cu surfaces, were linked by other groups to surface OH species through D₂O experiments and DFT calculations.⁴² Their DFT calculations propose that the 500 cm⁻¹ band corresponds to a top-site OH stretching mode, possibly aligning with our scenario. In contrast, the 700 cm⁻¹ band was attributed to the bending mode of free Cu-OH, a band mode that may be absent in our case due to co-adsorption with CO.”

2. Regarding the CO_{cov}, although authors refer to a recent literature, for clarifying to the potential readers, I think some extra information about how to estimate/calculate CO_{cov} from the Raman data, including the proper equation and the method to calculate it should be provided.

Response: We added more information and explanation of the estimation of the CO_{cov} to increase the understanding of the readers.

Changes in the manuscript:

Page 5, Line 115. “The relative CO surface coverage (CO_{cov}) can be obtained by the intensity ratio of the two CO bands as $\text{CO}_{\text{cov}} = \text{Intensity}(\text{CO}_s) / \text{Intensity}(\text{CO}_r)$. This relationship was derived in detail in our previous study through operando measurements in the presence of different CO concentrations and DFT vibrational analysis for different CO coverages on Cu(100) surfaces.³³”

3. I am concern about the reproducibility of the data. As it is well known, when working with plasmonic materials, there is the possibility of placing on/or nearby a hot-spot, which enhance the signal allowing to better follow the changes in Raman signal. A common practice when working in this field and, in order to give representative measurements, providing a similar behaviour in different parts of the electrode is recommended. Have the authors made sure that the signals are representative along the studied electrode? Could the authors provide some reproducibility data?

Response: We understand the concern of the reviewer. We have in fact repeated these experiments on different spots of the electrodes as well as on different electrode samples, where we always observed similar results. Additionally, we remind here that the measurements shown here are averaged from 20 pulse sequences that gives already a certain error in the data but also partially account for the issues described by the reviewer. We added examples of the three most important regions (EtOH, $\text{C}_2\text{H}_4\text{O}_x$ and C_1 region) from different freshly prepared electrodes, which show similar trends in the spectra and corresponding fits of the Raman bands. We added the Figure to the Supplementary Manuscript as well as commented on the reproducibility measurements in the Main Manuscript.

Changes in the manuscript:

Page 7, Line 193. To verify the reproducibility of the data, the experiments of the main product regimes were repeated on freshly prepared electrodes showing similar spectra and trends (Supplementary Fig. 9).

Changes in the supplementary manuscript

Page 10. Addition of Supplementary Figure 9 with repetition measurements of the characteristic pulse conditions.

Supplementary Figure 9. (a-c) Normalized SERS spectra on freshly individually prepared electrodes analogue to those in Fig. 2 but carried out on a different sample during pulsed CO_2RR with varying pulse lengths at $E_c = -1.0\text{V}$ and $E_a = +0.6\text{V}$ and (d-f) intensities of the fits of characteristic SERS bands averaged over one pulse sequence at selected pulse lengths. The red lines denote the exponential fits from Fig. 2 and serve as comparison. Despite some small differences arising from changes in the surface structure when comparing differently freshly prepared samples, these reproducibility measurements reported here still align very well with the results in Fig. 2.

4. In page 7, authors argue that “For the OH_{ad} species, in turn, it is not easy to extract the ad and desorption behavior since the bands additionally shifted upon the potential switch”. Although authors argue this, they do not show the extend for the change in the Raman shift. It is recommended to provide a plot with the change in the Raman shift to further support that the calculation of the ad and desorption is difficult to calculate. Related to this, why the shifting makes it difficult to calculate the ad and desorption rate constant?

Response: We have already shown the shift of the OH_{ad} band earlier in this study. On page 5, line 123 we introduced the OH_{ad} band with the following description:“(II) the OH_{ad} vibration on Cu, which can appear during the cathodic pulse at 495 cm⁻¹ and during the anodic pulse redshifted at 450 cm⁻¹ if OH is adsorbed or close to the Cu surface;³⁴⁻³⁶.” Moreover, Fig.2 a,b shows the shift of the OH_{ad} band clearly during the anodic and cathodic pulses. The difficulty to determine the adsorption and desorption kinetics and calculate the corresponding rates is based not solely on the apparent shift of the Raman band but also on the overlap of the OH_{ad} band with the simultaneously evolving comparably broad and intense Cu₂O bands. Furthermore, it is likely that Cu-OH is an intermediate of Cu₂O formation during the anodic pulse (as explained in the discussion part). We added these considerations to the main manuscript text.

Page 8, Line 217. “For the OH_{ad} species, in turn, it is not easy to extract the ad- and desorption behavior since the bands additionally shifted upon the potential switch and partially overlap with the evolving Cu₂O bands during the anodic pulse (Fig. 2.a,b). [...] Furthermore, it is likely that OH_{ad} is an intermediate of the Cu₂O formation during the anodic pulse.”

5. In the statement “If one follows the time-dependent evolution of these bands in the C1 regime (Supplementary Fig. 13, **now 14**), the contribution of the band at 410 cm⁻¹ was the highest in the first second...” This is not clearly shown in supplementary Fig. 13 (**now 14**), could the authors clarify this point?

Response: We wanted to indicate here that the band at 410 cm⁻¹ exhibits its maximum intensity at the initiation of the anodic pulse and subsequently diminishes in intensity. Additionally, relative to the other two Cu₂O bands, the contribution of the 410 cm⁻¹ band decreases throughout the anodic pulse. We add further clarity on this aspect in the manuscript.

Page 10, Line 244. “The relative intensity contribution of the band at 410 cm⁻¹ is stronger during longer cathodic pulses, which may have been influenced by the higher ratio of OH_{ad}.³⁶ On the other hand, the intensity of the 530 and 620 cm⁻¹ bands contributes stronger for shorter cathodic pulses. Analyzing the time-dependent evolution of these bands in the C₁ regime (Supplementary Fig. 14) revealed that the relative intensity of the band at 410 cm⁻¹ was the highest in the first second of the anodic pulse and its spectroscopic weight increased only slightly in the following. Also, the evolution of the intensity of the 620 cm⁻¹ band suggests a temporary maximum in the initial phase of the anodic pulse. In contrast, the intensity of the 530 cm⁻¹ band increased monotonously over the course of the anodic pulse.”

6. Page 12, line 283, for me, an increase in COO⁻ is not clear from the figure. Could authors clarify this point?

Response: The selectivity changes of COO⁻ are shown in Supplementary Fig. 18. We added this as reference in the manuscript text.

Page 14, Line 337. “[...] while the FEs of H₂ and C₁ products such as CO and COO⁻ (Supplementary Fig. 18a) increased.”

7. My concern about data shown in Figure 3 is related to the validity of the SERS values. It is well known that the intensity of a specific band depends not only on the quantity but also on the SERS enhancement at this specific moment that depends on the changes in the substrate. In other words, we should discard the reason for the value of the signals, whether they come from the amount of material or on the SERS enhancement. The authors explain very well that they used the CO_{cov} (from the ratio between COs/CO_r) to normalize the data. This is a very clever method; however, I think the validity of such normalization should be demonstrated. I would think that, in order to normalize the data, a spectator compound, which is a compound that is not participating in the reaction as an intermediate or something similar, should be more ideal. In this case, if the spectator increases the signal because of a SERS effect, this signal would serve as a normalization parameter (as it happens with an internal standard in analysis). Therefore, I am not sure about the validity of using CO_{cov} as a normalization parameter. I think authors should address this point since, from this data, most of conclusions of the authors are given in this work, so I think this is the key point of the article. I strongly recommend to include a demonstration of the validity of using CO_{cov} as such normalization parameter.

Response: In our previous study, we established the validity of CO_{cov} by introducing varying amounts of CO and applying the Langmuir model, along with DFT calculations (see Ref 33: Figures 3 and 4 and Figure S25). It proves challenging to identify a spectator species in this context, since it could be done erroneously, and it could in fact potentially impact the reaction itself. Moreover, we need to be clear that we are also limited in the intermediate determination by the temporal resolution of the Raman method, so we thus not completely discard that some of the intermediate assignments that we (and all other groups make), might be in fact spectator species and that key intermediates are missed if their surface residence time is shorter than our temporal resolution in Raman (here the fastest measurements are 0.2 s per spectrum).

Nonetheless here instead of utilizing CO_{cov} for normalization, we consistently employed ratios of bands within a single spectrum, as this approach ensures comparability in the Raman shift region. OH_{ad} is systematically compared to CO_{cov} to provide a relative understanding of their interactions. Alternatively, we could have used the carbonate bands at ~1060 cm⁻¹, which could be predominated by the carbonate ions in the electrolyte near the electrode surface. Although the protonation state of the carbonate might change following variations in the local OH⁻ concentration, the concentration of carbonate ions should be relatively constant. Unfortunately, the intensity of the carbonate bands varies strongly with electrode potential which might be due to the presence of the adsorbates decreasing the intensity compared to the non-catalytic conditions.

Minor changes

1. In the supporting info, Supplementary Figure 7 and Supplementary Figure 10 (**now 11**) are missing, so I lose some information when evaluating this data.

Response: We apologize for the absence of these figures in the submitted PDF version. It appears there may have been an issue with the upload to the Nature portal. They are accessible in the Word version, and we will ensure their inclusion in the upcoming upload also as PDF.

2. Page. 9, line 216, maybe authors refer to Supplementary Table 2 instead of Table 1?

Response: We appreciate you bringing this to our attention. The reference should be to Supplementary Table 2 in this context.

Changes in the manuscript:

Page 10, Line 263. Change of “Supplementary Table 1” to “Supplementary Table 2”.

3. In Supplementary Figure 4, 5, 6, and 11 there is no sense in including “Y axis” values. They should be similar to those in the main text of the manuscript.

Response: We have removed the Y-axis values in Supplementary Figure 4-8, 13 and 20 to be more consistent with the main text of the manuscript.

Changes in the supplementary manuscript:

Pages 5-20. Y-axis values have been removed in Supplementary Figure 4-8, 13 and 20.

4. Page 11. Line 254, correct the expression “Cu-oxide related bands due their strong”

Response: We have reformulated the sentence.

Changes in the manuscript:

Page 11, Line 294: “Moreover, even though we could not quantify the intensity of bands related to Cu oxides due their strong SERS sensitivity and missing normalization parameter, we can still follow the kinetic evolution of O_{ad} at short anodic pulse lengths and surface Cu_2O at long anodic pulse lengths (Supplementary Fig. 17).”

Reviewer #3 (Remarks to the Author):

The manuscript by Antonia Herzog and co-worker presents an extensively experimental study on pulsed CO₂ electroreduction to C1 and C2 species on copper electrodes. The authors provided very rich data to support their conclusion of the role of hydroxide surface coverages related to the faradaic efficiencies of different products. This should be a very interesting and important work to modulate the surface coverage to tuner the reaction products. I think it should be proper for publication on the Journal after they consider questions and comments as follows.

Response: We thank the Reviewer for the positive assessment of our work and the constructive feedback.

1. In this work the surface-enhanced Raman spectroscopy is used here, but the authors didn't provide any characterization of surface plasmon resonance in the present copper electrodes. Could they provide the related data for the optical property of the present electrode materials?

Response: We agree to the Reviewer that it is important to provide a proper characterization of the surface plasmon resonance to better understand the surface enhancement during operando Raman spectroscopy. We did not provide it in our original manuscript as we were rigorously comparing relative intensities of the detected Raman bands in a rather narrow wavenumber regime. Thereby, we aimed to ensure insights into the relative spectroscopic weights independently from the surface enhancement factors and thus, into the relative change in surface coverage with adsorbates.

Technically, we see difficulties to study the optical properties of the working catalyst by measuring the transmittance and reflectance to determine the absorbance under CO₂RR conditions. The reduced Cu₂O nanocubes transform to porous nanostructures. These transformed nanostructures are only metallic under applied electrode potential since any exposure to the electrolyte, the water, or the ambient leads to substantial (surface) oxidation. This process changes the optical properties considering e.g. the Kirkendall effect leading to porous CuO_x structures. Unfortunately, we do not have yet an experimental setup to measure in situ wavelength-dependent reflectance and transmittance of the electrodes under electrochemical conditions. This would be an interesting design for a future study of these systems.

Following the question of the reviewer, we included the changing conditions of the catalyst and its potential influence on the optical properties and surface plasmons under working conditions.

Changes in the manuscript:

Page 4, Line 104: "We note that the optical properties of the working catalyst are changing under potential pulse conditions as the reduced Cu₂O nanocubes transform to rough, porous nanostructures. Therefore, to account for different SERS enhancement at different pulse length conditions, the SERS bands in this study were only compared to each other within the same spectra to obtain intensity ratios (Supplementary Note 2)."

2. Page 7, lines 161-181, the authors discuss the low vibrational frequencies and higher vibrational frequencies. These Raman spectral peaks are contributed to the different adsorption site and C-C coupling. On the other hand, there is another possibility, which the Raman peaks arise from the

dipole-dipole coupling, as discussed in previous studies. So it is better to provide further interpretation on this point.

Response: We agree with the reviewer that dipole-dipole interactions should be considered as an origin for the Raman bands. In the case of Cu-CO, it has been shown in several studies (Phys. Chem. Chem. Phys., 2018,20, 25892-25900, J. Phys. Chem. C Nanomater. Interfaces. 2019 Apr 4; 123(13): 8112–8121) that CO adsorbs on Cu and we therefore attribute the Raman peaks mainly to adsorption. However, in the case of hydroxide, the Raman band either arises from absorption or dipole-dipole interactions. However, the molecules must be at least very close to the surface in order to experience the SERS effect. Therefore, even if a molecule is not directly adsorbed, there should be no other molecule adsorbed below it. We added the possibility of dipole-dipole interactions for hydroxide to the manuscript.

Changes in the manuscript:

Page 5, Line 111: “(I) the CO_{ad} vibrations on Cu, namely the Cu-CO rotation (CO_r) and stretching (CO_s) vibrations at 280 and 360 cm^{-1} , where CO is considered to be adsorbed on Cu as shown with surface characterization methods.^{31, 32} [...]. (II) the OH_{ad} vibration on or close to Cu, which can appear during the cathodic pulse at 495 cm^{-1} and during the anodic pulse redshifted at 450 cm^{-1} if OH is adsorbed or close to the Cu surface via dipole-dipole interactions.³⁴⁻³⁶”

3. Page 9, line 209, in the sentence “These changes might represent the formation/crystallization of the Cu_2O layer”, how to understand the surface can be thought as the crystallization of the Cu_2O layer? Could the authors provide further evidence?

Response: We admit that “crystallization” might be a misleading word. During the anodic potential pulse, we see the formation of Cu_2O in the Raman spectra. However, from our previous work we learned [Ref 18] that this Cu_2O crystallizes over the course of the anodic pulse. We identified Cu_2O like Bragg peaks in the operando high-energy X-ray diffraction pattern after ~1 s. Within the 10 s anodic pulse, we could follow the growth of the Cu_2O domains, likely as a termination layer on the Cu domains, up to 3 nm. We changed this section in the manuscript accordingly.

Changes in the manuscript:

Page 10, Line 252: “These changes might represent the formation/crystallization of the Cu_2O layer terminating the Cu domains. In our previous work, we showed with operando X-ray diffraction that over the course of an anodic potential pulse, crystalline Cu_2O domains with a size of up to 3 nm were present at the end of a 10 s anodic pulse.¹⁸ Additionally, operando X-ray absorption spectroscopy revealed a minor contribution of other Cu phases (e.g., Cu(II) compounds such as $\text{CuO}/\text{Cu}(\text{OH})_2$) in the initial phase of the electrochemical oxidation which are likely highly disordered and thus absent in the XRD data.”

4. Page 11, lines 248-252, when the discuss the kinetics of the adsorption and desorption of COs and CO_r vibrations, they thought the surface coverage of adsorbed CO, but it is not clear what/how much is the surface coverage of adsorbed CO on the copper surface?

Response: Here we would like to refer to our previous Raman work in which we present how the intensity ratio of the two CO bands as $CO_{cov} = \text{Intensity}(CO_s) / \text{Intensity}(CO_r)$ could be linked to the CO coverage [Ref 33]. This relationship was derived through operando measurements in the presence of different CO concentrations and DFT vibrational analysis for different CO coverages on Cu(100) surfaces. However, the structural and morphological differences between the Cu(100) derived relationship to the oxide-derived Cu used in this work does not allow a direct comparison of the ratios and the CO coverage. Thus, we unfortunately cannot make any quantitative statements on the CO coverage on the Cu surface during (pulsed) CO₂RR. We tried to make the origin of the CO coverage clearer in our manuscript.

Changes in the manuscript:

Page 5, Line 115. “The relative CO surface coverage (CO_{cov}) can be obtained by the intensity ratio of the two CO bands as $CO_{cov} = \text{Intensity}(CO_s) / \text{Intensity}(CO_r)$. This relationship was derived in detail in our previous study through operando measurements in the presence of different CO concentrations and through DFT vibrational analysis for different CO coverages on Cu(100) surfaces.³³ However, the structural and morphological differences between the Cu(100) derived relationship to the oxide-derived Cu used in this work does not allow a direct comparison of the ratios and the CO coverage.”

5. Page 11, line 262, Figure 4 and Supplementary Figure 17 (**now 18**), these authors discuss the influence of OH adsorption on the change of Faradaic efficiencies different reduced species. It is quite strange why ethylene has so large change in this Faradaic efficiency in the wide range of OHad/Coad ratio? This becomes especially significant in the lower and higher regimes. They should present further comments for this point.

Response: We believe that the decrease of the ethylene Faradaic efficiency mainly results from the irreversible morphology changes caused by the pulsed reaction conditions (see ex situ SEM images in Supplementary Fig. 21), which are especially prominent at longer anodic pulse length durations. We have discussed the behavior of the ethylene Faradaic efficiency later in our discussion part on page 15. To make it easier for the reader we added a comment to Figure 4 in the Manuscript.

Changes in the manuscript:

Page 14, Line 341. “We note that the decrease of the ethylene FE may be predominantly linked to irreversible (morphological) catalyst changes during the harsh working conditions as shown in our previous work and will be further discussed later.¹⁸”

6. In the reference list, line 607, reference 45 (**now 48**) should be corrected.

Response: We corrected reference 45 (**now 48**).

REVIEWERS' COMMENTS

Reviewer #1 (Remarks to the Author):

The authors have thoroughly addressed my concerns. I recommend publication of the revised manuscript in its present form.

Reviewer #2 (Remarks to the Author):

In relation to the revised paper, I consider that the authors have made a thorough revision of the manuscript, substantially improving those parts that were of concern in the first version. Therefore, I consider that the paper is suitable for publication in Nature Communications.

Reviewer #3 (Remarks to the Author):

There is the reference number 48 (page 26) of the reference list which is not in the complete form.